# The Online HEARRING Counselling 1.0 Platform Provides Clinicians with Comprehensive Information on Hearing Device Solutions for Conductive, Mixed, and Sensorineural Hearing Loss

**DOI:** 10.3390/jpm12122027

**Published:** 2022-12-07

**Authors:** Rudolf Hagen, Kristen Rak, Anja Kurz, Wolf-Dieter Baumgartner, Javier Gavilán, Paul van de Heyning

**Affiliations:** 1The Comprehensive Hearing Center, Department of Oto-Rhino-Laryngology, Plastic, Aesthetic and Reconstructive Head and Neck Surgery, University of Wuerzburg, Josef-Schneider-Straße 11, 97080 Würzburg, Germany; 2ENT Department, Vienna ENT University Hospital, Spitalgasse 23, 1090 Vienna, Austria; 3Department of Otolaryngology, IdiPAZ Research Institute, La Paz University Hospital, Paseo de la Castellana, 262, 28046 Madrid, Spain; 4ENT Department, Antwerp University Hospital, Wilrijkstraat 10, 2650 Antwerp, Belgium

**Keywords:** hearing health care counselling, World Café, single-sided deafness, bone conduction, HEARRING, hearing candidates

## Abstract

A platform to help clinicians ensure that hearing device candidates are informed about the benefits and drawbacks of their recommended treatment option would be of clinical counselling benefit because it could help each candidate form realistic expectations about life with their treatment option. Following the World Café approach, 54 participants (surgeons, audiologist, and researchers) generated lists of the benefits and drawbacks of each treatment option for single-sided deafness (SSD) and bone conduction (BC) solutions. They then prioritized the benefits and drawbacks. After the World Café, literature research was performed on each topic to check if the statements (on benefits and drawbacks) are supported by quality peer-reviewed publications. Each participant was surveyed to ensure a collective agreement was reached. The HEARRING Counselling 1.0 Platform was developed. Thus far, sections for SSD and BC solutions have been completed. Initial feedback has been highly positive. The platform will be expanded to cover middle ear implant solutions and cochlear implants. A plan is in place to ensure the information continues to be timely. The HEARRING Counselling 1.0 helps clinicians provide comprehensive information to candidates about their treatment option and thereby helps establish that candidates have realistic expectations about the benefits and drawbacks of device use.

## 1. Introduction

The interventions currently best able to mitigate the effects of hearing loss are hearing devices, e.g., hearing aids (HA), middle-ear implants (MEI), bone conduction devices (BCD), cochlear implants (CI), and auditory brainstem implants (ABI) [1,2,3,4,5,6,7]. While these devices are often for different types and severities of hearing loss, they do sometimes share indications, e.g., CI, BCD, and contralateral routing of signal hearing aids (CROS-HA) for people with single-sided deafness (SSD) [8,9]. A guide for the advantages and disadvantages of each type of intervention, and indeed of the choice of “no intervention”, according to different types of hearing loss would therefore be of practical clinical use when counselling device candidates about their recommended treatment option. Such a guide would provide relevant information and thereby, hopefully, be a step towards each candidate receiving expert information on the benefits and drawbacks of using the recommended device best suited to their needs.

There are, however, no standardized procedures or international good practice guidelines for informing device candidates about the benefits and drawbacks regarding their recommended treatment option. By providing such guidelines, a globally accepted basis to ensure each candidate has realistic expectations about life with their device could also be established. To this end, the authors set the goal of creating a platform that should (1) detail the evidence-based benefits and drawbacks of each intervention, (2) not be specific to any one manufacturer, and (3) not be country- or region-specific. Thus, the primary aim of this article is to present the developmental process and first results with the HEARRING Counselling 1.0 platform.

## 2. Materials and Methods

### 2.1. Contributors

The contributors were members of the HEARRING group, which is an independent network of world leading centers and experts dealing with all aspects of hearing disorders. They believe that advancements in the field of hearing devices are achieved through international peer-reviewed research and the pooling of collective experience. Therefore, HEARRING members are committed to leading the research in hearing device science, advancing audiological procedures, and developing and perfecting surgical techniques.

In May 2019 the Steering Committee of the HEARRING group defined the goal of finding a way to standardize the clinical counselling process regarding hearing loss treatment. Invitations were sent to all HEARRING centers. 54 individuals (surgeons, audiologists or researchers), all with advanced experience in hearing health care, agreed to participate.

A World Café Meeting took place on 5–7 November 2019, hosted by the University of North Carolina (Chapel Hill, NC, USA). During this meeting the group primarily aimed to assess benefits and drawbacks users should be informed about before deciding on a treatment regarding BCD and SSD. Additional meetings will take place to discuss conductive, mixed, and sensorineural hearing loss (i.e., MEIs) and for severe to profound sensorineural hearing loss (i.e., CIs).

### 2.2. The World Café Method

The World Café approach is a well-established method to facilitate discussion amongst large groups of people. Via a succession of structured conversations, participants can work collaboratively towards a greater understanding of the questions at hand [10,11]. It was used by health care providers for such diverse ends as creating healthier communities for rural, older adults living with diabetes; designing educational initiatives; discussing the efficiency of African swine fever control strategies in European wild boar population; and discussing the need for community-based stroke recovery [12,13,14,15]. The present paper is the first instance, to the best of our knowledge, of the World Café approach being used in hearing health care.

In accordance with the method established by the World Café Community Foundation [11], the 54 participants (surgeons and audiologists) were divided into 2 rooms (1 for BCD and 1 for SSD) of 27 participants each. In each room, the participants were further divided into 4 subgroups, each at their own table (6–7 people per table). It was ensured that each group was comprised of both surgeons and audiologists. As can be seen in Figure 1, each of the 4 tables was for a specific topic and had tasks to complete. After 8 min of discussion at the selected starting table, the groups were then rotated to the subsequent table. This method was performed until each group had completed a full rotation.

Each table during the discussion rounds was led by a facilitator, whose role was to encourage discussions and all participants’ active participation without influencing the content. Every participant could add points, prioritize statements, discuss his/her thoughts, and draw schemes on the flip charts. All data were collected, summarized, and presented on the same day. Facilitators stayed at their table and did not rotate with the group. The groups did not change rooms; therefore, the 27 participants that started in the BC room did not discuss the SSD afterwards and vice versa.

### 2.3. Supporting the Results with Literature

To add weight to each statement, literature research was performed on each topic to check if the statements are supported by peer-reviewed publications. This was performed by 4 HEARRING centers: The ENT Department of Antwerp University Hospital (Antwerp, Belgium), Madrid Hospital La Paz, Vienna ENT University Hospital (Madrid, Spain), and Würzburg ENT University Hospital (Würzburg, Germany). Statements which contradicted the scientific consensus would be deleted from the list, if necessary. To show direct evidence of the information, direct or indirect citations as well as consensus statements of those publications were assigned to each statement. This comprehensive amount of data was reviewed as a draft by the same 4 HEARRING centers.

Two surveys were distributed by the center in Würzburg to give all HEARRING members the opportunity to check or suggest changes to all statements, prioritizations, and the literature of all subgroups. This enabled members of the BC group to review and comment upon the results of the SSD group, and vice versa. Additionally, participants were again asked to prioritize each statement to get a collective opinion on each topic. To check the comprehensiveness of the publications selected, each member could also add or delete citations and publications from the list.

The benefits and drawbacks of each intervention was assigned a “HEARRING Importance Factor”. This is a 1–5 scale in which 5 indicated it is very important for every user and 1 indicated it is highly important to only a subgroup of users. This rating was based on the prioritization established by the 3rd and 4th tables in the World Café meeting and was finalized after review by all participants.

Finally, the results of the surveys were collected and summarized.

## 3. Results

During this data cleansing process, a visual and dynamic platform was developed to make the information more accessible and transparent. The outcome is presented in the HEARRING Counselling 1.0 Platform [16]. (Figure 2)

To navigate the platform, the user clicks a treatment class, e.g., single-sided deafness. This opens onto a page that presents the clinical options: no treatment, CROS hearing aids, bone conduction devices, and a cochlear implant. For each option, the benefits and drawbacks of this type of intervention (or lack of intervention) are categorized into medical aspects, audiological aspects, and users’ reports. By clicking on a statement, a list of quotations from important peer-reviewed publications underline the statement and put it into context. Furthermore, links to peer-reviewed publications provide the opportunity to aid those who wish to do more research about the topic. The importance of each benefit and drawback (the “HEARRING importance factor”) is shown in the middle of the screen (Figure 3).

When complete, the platform will contain four sections: (1) solutions for single-sided deafness (to show state-of-the-art treatment options for SSD), (2) bone conduction solutions (to treat conductive and mixed hearing loss), (3) middle ear implant solutions (to treat conductive, mixed and sensorineural hearing loss), and (4) cochlear implant solutions (for bilateral severe to profound sensorineural hearing loss). At the time of writing, sections for SSD and BCD are complete and sections for MEI solutions and CI solutions are in development.

Further, there are two tabs on the homepage: one to the HEARRING quality standards and one to a list all the references used in the counselling platform. The former were included because the HEARRING group recommends that before starting a program, e.g., a bone conduction program, clinics (and therefore their patients) would benefit from implementing the applicable quality standards.

To ensure that the information in the platform keeps pace with new research, the content will be updated every 12 months. To do this, one HEARRING center will perform a literature search, select appropriate statements, and propose a HEARRING importance factor to each. A survey to all HEARRING members will ensure the peer-reviewed process during this update. At the same time, already available content shall be reviewed in order to guarantee up-to-date information according to best clinical practice.

The current contents of the counselling platform were integrated into clinical routine by the HEARRING group in Würzburg using a computerized program. At the time of writing, more than 200 patients were counselled using this standardized procedure. Users and patient feedback have demonstrated that the platform is comprehensive and easy to use and that the well-balanced information on advantages and drawbacks enables an objective counselling process.

## 4. Discussion

In this present study, a platform to provide clinicians worldwide with a timely and expert summary of the benefits and drawbacks of each type of intervention was presented. Having this information at hand should be useful for clinicians to help candidates gain realistic expectations regarding device use, thereby creating a win-win situation for clinician and candidate. The platform is free to use, is not specific to the devices of any one company, and lists and assigns various levels of importance to the benefits and drawbacks of each intervention and supports them with quotations from published peer-reviewed papers. The links to these papers are also provided in the platform. The selection of the benefits and drawbacks, and the literature curated to support them, were the result of a lengthy expert-driven process. As such, the platform can function as a comprehensive “cheat sheet” of information for clinicians to support comprehensive counselling.

While the quotations in the platform are written at the clinician- rather than lay-level (since they come from scientific papers), they do nonetheless state the expected outcomes of device use. As such, clinicians could use the platform to explain in lay language the benefits and drawbacks to interested candidates. Indeed, making sure that device candidates have realistic expectations is something clinics strive to achieve and, in studies in which participants receive an implant, is often an inclusion criterion [17,18,19]. Realistic expectations cover outcomes and the level of commitment needed to reach such outcomes. It remains the responsibility of the clinicians to ensure the recipient understands the latter.

For further guidance/suggestions on pre-intervention counselling, clinicians may benefit from reading the applicable sections in the various HEARRING Quality Standard publications (e.g., [20,21]). These standards, which present standards that are a realistic minimum attainable by all implant clinics, are reachable via the tab on the homepage of the platform, as can be seen in Figure 2.

It is important to note that the platform never recommends a specific intervention. It is not interactive or adjustable, so it does not factor in worldwide variation in indications/contraindications, etiology, level of patient motivation/commitment or family support, if devices are approved in the clinician’s/patient’s country, or reimbursement status [22,23]. It is the clinician’s responsibility to use this information provided in the platform in conjunction with the real-life conditions in their country to find the most suitable solution for each candidate. As a result, the platform has the advantage of being applicable worldwide.

The real-life factors, such as reimbursement, while not a part of this platform, are nonetheless of great importance. Hopefully the use of this platform will not only help understanding benefits and drawbacks of each device category, but also create awareness about a whole portfolio of possible solutions, e.g., treating SSD and conductive and mixed hearing losses via BC devices. Only if all treatment possibilities are reimbursed, clinicians can counsel comprehensively. It is our hope that this may play a part in leading more health insurance providers to reimburse more devices thereby making them realistically available to candidates; only in that way can people receive an optimally individualized treatment. While this platform is not a substitute for a cost-benefit analysis, it can provide something like a generalized expert consensus/summary on expected outcomes with each intervention.

Other potential uses of the platform could be as a teaching aid or as a convenient reference to for clinicians to consult during clinical practice.

One limitation of the platform is that it is only available in English. Its impact would certainly be greater if it were available in a variety of languages; however, as version 1.0. is not yet finished, this consideration was postponed.

## 5. Conclusions

The HEARRING Counselling 1.0 Platform condenses the many years of experience of the worldwide HEARRING centers and was shown to improve the quality of the clinical counselling process. It provides clinicians with easy to access, up-to-date evidence-based information on the benefits and drawbacks of different kinds of hearing devices. It is universal in that it is not specific to any single manufacturer or any specific region in the world. This platform helps clinicians provide comprehensive information to candidates about what they can expect once the best treatment option was determined and thereby more firmly establish that candidates have realistic expectations about the benefits and drawbacks of device use. We hope that this evidence-based approach will be a part of broadening reimbursement possibilities for more devices and in more nations; thereby enabling as many people as possible to benefit from their best possible hearing. We hope that experience will show that clinicians find the platform useful and that it can be translated into their local languages.

## Figures and Tables

**Figure 1 jpm-12-02027-f001:**
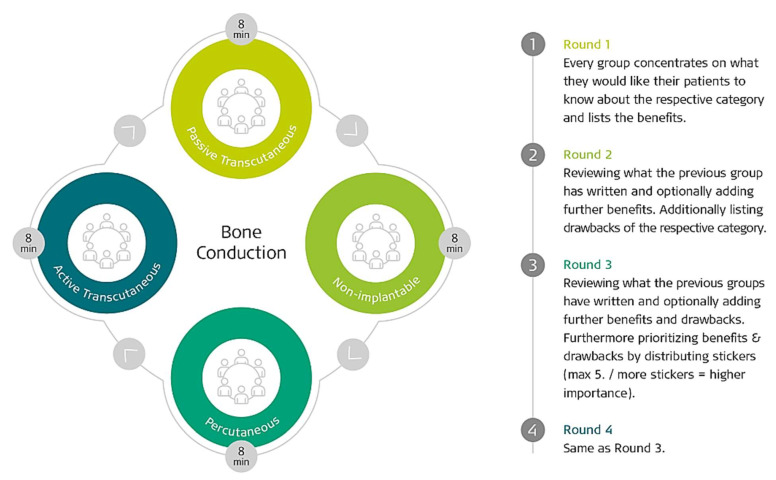
The four tables and how participants rotated. In this example they were discussing bone conduction solutions.

**Figure 2 jpm-12-02027-f002:**
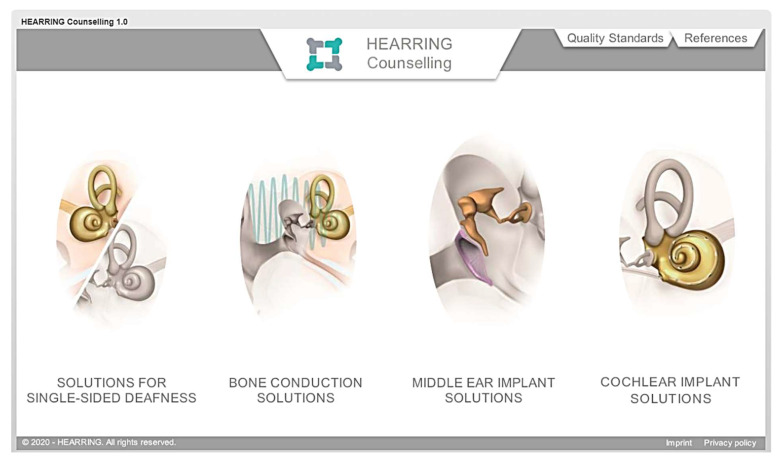
The home screen of the HEARRING counselling platform 1.0.

**Figure 3 jpm-12-02027-f003:**
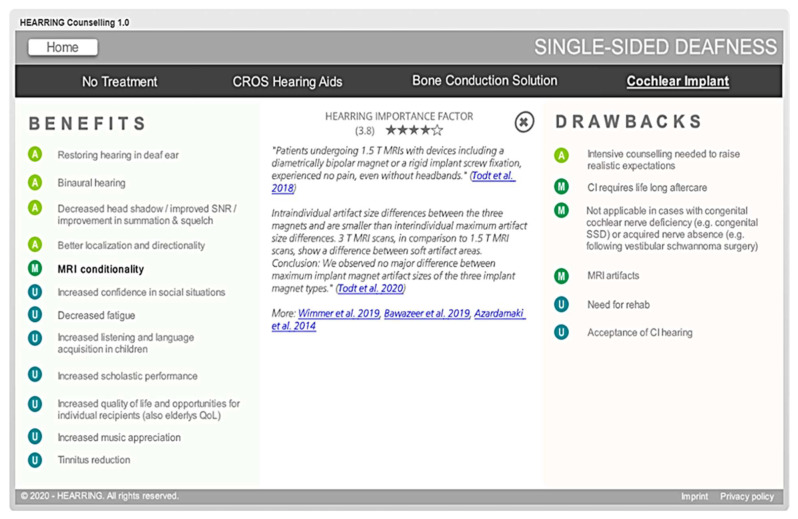
Inside one of the treatment options, here for single-sided deafness. In this screenshot, cochlear implants and the benefit “MRI conditionality” is displayed.

## Data Availability

Not applicable.

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
