# Peer review of "The Online HEARRING Counselling 1.0 Platform Provides Clinicians with Comprehensive Information on Hearing Device Solutions for Conductive, Mixed, and Sensorineural Hearing Loss"

_jpm, 2022, doi:10.3390/jpm12122027_

Round 1

Reviewer 1 Report

The manuscript describes a computer-based "platform" intended to provide information about positive and negative aspects of potential treatment options.  The manuscript provides a thorough account of how the platform was developed, with suggestions and discussion from a range of experts evaluated in a qualitative but systematic way.  It also provides information about how the platform could be used by clinicians, although in less detail.  Also, there is no information about whether clinicians will find this to be valuable.  It will collect and display information in a single place, which will surely be worth something.  Some aspects, like updating only once a year, seem not to take the greatest advantage of what is possible in a modern online database.  The report feels preliminary, but perhaps that is because, as noted at line 209, at version 1.0, this is not yet a finished product.

Author Response

  • There is information on clinicians finding this valuable. Please see lines 160-165:

“The current contents of the counselling platform were integrated into clinical routine by the HEARRING group in Würzburg using a computerized program. As of the time of writing, more than 200 patients have been counselled using this standardized procedure. Users and patient feedback have demonstrated that the platform is comprehensive and easy to use and that the well-balanced information on advantages and drawbacks enables an objective counselling process.”

  • Certainly, updating it more often than once per year would be ideal; however, reaching the consensus with so many people on the updates is a time-consuming affair. Hopefully, the 1x a year will be the minimum and not the standard update time.
  • It is preliminary, in the sense that 1) the tool is not finished and that there is not a lot of feedback from clinicians (especially those that don’t work at the clinic primarily responsible for creating the tool), which is reasonable because the platform is new. It’s quite possible that once it’s used by more clinics, their feedback will help us create a better platform and 2) that 2 of the 4 solutions (middle ear implants and cochlear implants) have not been completed it.

However, it is complete in the sense that the paper explains the purpose and hoped-for effects of the platform, and those will not change with the addition of the sections for middle ear implants and CIs. We don’t foresee that this will lead to the platform’s purpose expanding but admit that that is always possible.

Reviewer 2 Report

I generally enjoyed reading the manuscript. I propose the following points to conduct to strengthen it:

-         - Lines 74-76: When the authors mention that The World Café method is a well-established method to facilitate discussion amongst large groups of people, they provide two references (10 and 11). I suggest elaborating a list of references here, to add more specific references, directly related to the similar topic/field.

-         -  Lines 81-82: how were surgeons and audiologists classified into groups? Randomly? Do we have more background data for them? Experiences, years of work etc. How were both groups balanced?

-          Line 108: Please, add the survey into appendix. Could the authors list variables somewhere?

-          - In the conclusions: please, add practical and theoretical implications, limitations and future outlook.

-          - I believe that the manuscript would benefit from adding and discussing the following work: Debevc et al. Effectiveness of a self-fitting tool for user-driven fitting of hearing aids. International journal of environmental research and public health, 2021.
